# Parietal Alpha Oscillations: Cognitive Load and Mental Toughness

**DOI:** 10.3390/brainsci12091135

**Published:** 2022-08-26

**Authors:** Natalia Zhozhikashvili, Ilya Zakharov, Victoria Ismatullina, Inna Feklicheva, Sergey Malykh, Marie Arsalidou

**Affiliations:** 1Neuropsy Lab, Faculty of Social Sciences, HSE University, 101000 Moscow, Russia; 2Developmental Behavioral Genetics Lab, Psychological Institute of Russian Academy of Education, 125009 Moscow, Russia; 3Laboratory of Molecular Genetic Research of Human Health and Development, Scientific and Educational Center “Biomedical Technologies”, Higher Medical and Biological School, South Ural State University, 454080 Chelyabinsk, Russia; 4Department of Psychology, Lomonosov Moscow State University, 119991 Moscow, Russia; 5Faculty of Graduate Studies, York University, Toronto, ON M3J 1P3, Canada

**Keywords:** alpha oscillations, effort, encoding, mental toughness, recognition, retention, task difficulty, working memory

## Abstract

Cognitive effort is intrinsically linked to task difficulty, intelligence, and mental toughness. Intelligence reflects an individual’s cognitive aptitude, whereas mental toughness (MT) reflects an individual’s resilience in pursuing success. Research shows that parietal alpha oscillations are associated with changes in task difficulty. Critically, it remains unclear whether parietal alpha oscillations are modulated by intelligence and MT as a personality trait. We examined event-related (de)synchronization (ERD/ERS) of alpha oscillations associated with encoding, retention, and recognition in the Sternberg task in relation to intelligence and mental toughness. Eighty participants completed the Sternberg task with 3, 4, 5 and 6 digits, Raven Standard Progressive Matrices test and an MT questionnaire. A positive dependence on difficulty was observed for all studied oscillatory effects (t = −8.497, *p* < 0.001; t = 2.806, *p* < 0.005; t = −2.103, *p* < 0.05). The influence of Raven intelligence was observed for encoding-related alpha ERD (t = −2.02, *p* = 0.049). The influence of MT was observed only for difficult conditions in recognition-related alpha ERD (t = −3.282, *p* < 0.005). Findings indicate that the modulation of alpha rhythm related to encoding, retention and recognition may be interpreted as correlates of cognitive effort modulation. Specifically, results suggest that effort related to encoding depends on intelligence, whereas recognition-related effort level depends on mental toughness.

## 1. Introduction

Mental toughness is a personality trait that enables individuals to cope better with challenging tasks. Behavioral research shows that individuals who score high on mental toughness have higher productivity [1], academic performance [2] and performance on cognitive tasks [1,3,4,5,6]. It is believed that people with high mental toughness scores invest more effort in cognitive tasks by encoding relevant information and inhibiting irrelevant information, which leads to better performance [7]. According to neuroimaging studies, encoding relevant and inhibiting irrelevant information is reflected in desynchronization and synchronization in the parietal alpha rhythm, respectively [8,9,10]. Individual variability of desynchronization and synchronization in parietal alpha rhythms may be related to emotional and motivational processes related to task difficulty, as a consequence, expressed by a varying degree of effort [11,12]. No study to date has investigated interrelations among brain correlates of cognitive processes, intelligence and mental toughness. The present electroencephalography (EEG) study investigates how variability in mental toughness and intelligence relates to parietal alpha oscillations recorded during working memory processes (i.e., encoding, retention, retrieval) across four levels of difficulty. The results of our study will improve understanding of the role of personality characteristics in the performance of various cognitive processes and advance knowledge on the cognitive mechanism for the distribution of effort between different cognitive processes.

Alpha oscillations (8–13 Hz) are a dominant (peak) frequency in adults [9]. Alpha power is thought to be the inverse correlate of brain activation: inhibition of alpha power is interpreted as the arousal of the cortex, during which a variety of cognitive processes are required to perform various complex cognitive tasks [10,13,14,15]. The degree of alpha power suppression is positively correlated with indicators of brain activation obtained using functional magnetic resonance imaging [16,17]. Different opinions exist about the functional significance of the alpha rhythm. Some authors suggest that EEG alpha oscillations do not have any functional significance and reflect a biological artifact [18], whereas others propose that the alpha-rhythm is a marker of cognitive inactivity or generalized idling brain state [19,20]. More recent hypotheses relate the alpha rhythm with inhibitory control and redistribution of neural resources [21].

Stimulus-related alpha power suppression relative to a prestimulus baseline is termed event-related desynchronization (ERD). ERD is observed in a wide variety of cognitive tasks that require perceiving a stimulus such as mental arithmetic [22,23], reading [24,25], task conditions engaging specific attention and memory processes [26,27,28,29], memorization tasks [30], and working memory [31,32]. Notably, most studies report ERD in the parietal cortex. The parietal cortex serves a crucial role in transforming sensory input into motor output and is involved in many cognitive operations, including spatial representation, attention, sensorimotor transformation and abstract planning [33]. Research shows that the lower the alpha power associated with a stimulus, the higher the accuracy of the task [9,10]. There is a general agreement that alpha desynchronization reflects cortical activation [10] and is directly related to the expenditure of resources necessary to complete a cognitive task [34]. Further, ERD increases with the complexity of the task, as more complex tasks require more cognitive resources [9,34,35]. This effect reveals the relation of alpha desynchronization with attentional demand and may reflect an increase in sustained attention and in the amount of allocated cortical resources [30], which may be driven by activation of thalamocortical networks associated with memory processes [36]. Some authors suggest that alpha desynchronization reflects the amount of encoded and retrieved information that is parametrically related to the decrease in synchronized firing of neurons [37,38,39]. Researchers have also interpreted this effect as an increase in effort invested in task performance [40,41]. Overall, parietal alpha ERD is associated with cortical activation reflecting expenditure of cognitive resources.

Event-related synchronization (ERS) was the focus of fewer EEG studies. ERS reflects an increase in alpha power. It occurs when a task requires inhibiting a response [10], ignoring irrelevant distractors [42] or retention of information in working memory [8,43]. Researchers interpret ERS as inhibition or inhibitory control, which allows one to slow down cognitive processes and ignore irrelevant information [8,10]. Some suggest that alpha synchronization reflects inhibition of task-irrelevant regions of the neuronal network that allows information to gate into task-relevant brain areas [10,44]. Moreover, when performing tasks that require information retention, alpha synchronization is interpreted as active processing associated with memory maintenance [45]. This phenomenon also increases with the complexity of the task [46,47], which may reflect the ‘idling’ of this frequency band or increased top-down functional inhibition of brain regions that can possibly perturb maintenance when attentional demands are higher [9,47]. Thus, ERS is associated with inhibitory processes needed during problem solving.

In the current study, we examine ERS and ERD alpha correlates of the Sternberg task [48]. The Sternberg task is a popular working memory task. It requires participants to encode a set of items presented on the screen, retain them for a short period of time, and in a subsequent screen identify whether a target item was a member of the original set. Neuroimaging studies show that these three stages of performance are expressed in the modulation of the alpha rhythm [8,43,49,50]. Posterior ERD is observed during the stage of encoding information (ERD during encoding), posterior ERS is observed during the stage of information retention (ERS during retention), and posterior ERD is observed again before and during the response phase (ERD related to response preparation and execution). ERD related to encoding [51] and retrieving [46] and ERS related to retention [46,47,49] increases with task difficulty. Critically, no study has revealed in the same sample both effects when examining Sternberg task performance. Specifically, studies that examined both these effects did not reveal significant relations between encoding-related ERD and task difficulty [8,46,47,49,50]. Further, most studies do not calculate the individual alpha frequency for their analyses, while alpha frequency peak is known to show large individual differences and, thus, analysis of fixed frequency ranges may not be useful for studying alpha power modulation effects [9]. Our study examines the first-time effects of ERD/ERS across four levels of difficulty and across three cognitive processes (encoding, retention and responding). The main goal of our study is to identify how ERD/ERS related to difficulty and cognitive process are modulated by individual differences in mental toughness and intelligence.

Individual differences correspond to variation in traits and characteristics among individuals. Intelligence has been shown to be an important factor for task performance in numerous EEG studies [52,53]. Some studies showed that people with high intelligence demonstrate lower ERD when performing a task [11,54,55]. The Neural Efficiency hypothesis proposes that participants with higher intelligence are more efficient and thus can complete a task with less resources [56]. However, other studies observe the opposite effect, showing that ERD is increased in individuals with high intelligence when performing a task [57,58,59]. Authors often suggest that highly intelligent participants have access to more resources in solving challenging tasks. Differences in parietal cortex activation (expressed in both ERD; [57] and in blood oxygenated level-dependent signal [55,60]) for individuals who scored high and low on intelligence tests are proposed to vary by difficulty level. Specifically, the neural efficiency effect is observed in easy tasks, whereas for difficult tasks that require higher cognitive effort, the opposite effect is observed [56].

Analysis of brain activity as a function of performance scores (i.e., subjective) rather than a priori (i.e., objective) levels of difficulty also demonstrated fluctuations as a function of tasks that were experienced as easy or difficult [61]. Specifically, brain activity increased in highly intelligent participants and decreased in less intelligent participants when the task became difficult. Since subjective task difficulty was analyzed, it was suggested that the effect was driven by an individual’s reaction to subjective difficulty rather than by difficulty level. The authors suggested that this dependency should be explained by the level of motivation to perform difficult tasks, which differs between high-performing and control participants [56,61]. This assumption is also consistent with findings showing a trend towards a decrease in cortical activation at the most difficult levels for less intelligent participants [57], which is unlikely to be explained by the achieved ceiling effect. Critically, as studies do not typically use tasks with various levels of difficulty this effect has not been systematically investigated using EEG. Our study will examine behavioral and brain indices across four levels of task difficulty.

According to the Expected Value of Control Theory of mental effort [62], the resources allowed to activate the top-down control processes are limited, which requires their economy and rational use. Therefore, instead of performing a task at the maximum level of their capabilities, a person increases the efforts invested in completing a task as its requirements increase. According to the Motivational Intensity Theory the amount of effort invested, and therefore the amount of resources spent depends not only on the difficulty of the task but also on the motivation to complete it [12,63].

Many studies that examined the relationship between alpha power and task difficulty [11,54,55,57,58,59] did not control for motivational factors. Without considering personality traits associated with motivation, it is difficult to determine whether the modulation of alpha activity as an effort indicator is associated with emotional-motivational aspects or with merely the cognitive capabilities of participants. Mental toughness is a general term for a set of positive psychological characteristics that play an important role in individual achievement in various fields of activity as well as mental health [5]. One of the most popular mental toughness models characterizes mental toughness as a set of four interrelated but independent components: (1) control: a person’s tendency to feel and act as if one has an influence on the world around one’s life and the ability to maintain the level of anxiety is under control; (2) commitment: a tendency to actively and persistently strive for goals despite the difficulties that arise; (3) challenge: the tendency to view potential failures as opportunities for self-development and to continue to strive for desired changes; and (4) confidence in one’s abilities and in interpersonal relationships: maintaining high self-esteem of one’s personality and abilities despite setbacks and the ability to develop social bonds [64]. A mental toughness scale was derived as a factor of individual differences that allows people to effectively cope with life difficulties (such as negative life events, stressful situations, and failure in activity) and persistently pursue their goals [5]. Research shows that mental toughness correlates with motivation and goal orientation [65,66]. Moreover, mental toughness is positively associated with the level of productivity of employees with a high level of perceived stress, as well as with academic performance [1,3]. Thus, mental toughness reflects the set of personality characteristics necessary to maintain a sufficient level of effort when difficult tasks arise [5]. We hypothesize that when solving relatively easy tasks, the variability in alpha power related to task performance should be explained by intelligence, and when performing difficult tasks, this variability should be explained by the motivational indicators associated with the personality’s reaction to subjective difficulty (i.e., mental toughness). We hypothesize that the influence of motivational aspects as assessed by mental toughness is not driven by intelligence. Importantly, according to our knowledge, no study has dissociated the impact of individual differences associated with mental toughness and intelligence on the relationship between alpha power and task difficulty.

The current study examines for the first time the influence of mental toughness and intelligence on the relation between parietal alpha rhythm responses to three phases (encoding, retention and responding) and four difficulty levels of the Sternberg task. Specifically, we test the following main hypotheses: (1) ERD during encoding, ERS during retention, ERD during recognition are related to task accuracy; (2) ERD during encoding, ERS during retention, ERD during recognition increase with Sternberg task difficulty; (3) Only in easy task conditions, Raven scores will correlate with the alpha power effects; (4) Only in difficult task conditions, mental toughness scores will correlate with alpha power effects.

## 2. Materials and Methods

### 2.1. Participants

Eighty healthy right-handed adult participated in the study (20 ± 3.5 years (mean ± standard deviation; 49 females). The sample size was determined taking into account the previous EEG studies described in the Introduction, as well as the known understated power effect in EEG studies [67]. All participants had normal or corrected-to-normal vision; they reported no history of neurological, or mental disorders. The experiments were approved by the ethics committee of the Psychological Institute of Russian Academy of Education and signed informed consent was obtained from all participants. All experiments were conducted in a sound-attenuated chamber.

### 2.2. Materials

Participants completed a numeric Sternberg task with four levels of difficulty during EEG recording (Figure 1). The stimuli were presented and responses were recorded using Psychopy2 [68]. Each trial began with the presentation of a cross cue symbol (1 s duration). A total of 200 ms after the disappearance of the cross, a list of a unique combination of randomly chosen digits appeared (encoding phase; 1.5 s duration), followed by a blank screen retention interval (retention phase; 2 s duration), and then a target digit (retrieval phase; 2 s duration). Participants were asked to indicate whether a target digit was in the previous encoding set by pressing a button to respond yes or no. The time participants had to respond was not limited. The probe was in-set for 50% of all trials, and the order of in-set/out-set trials was pseudorandomized. There were four pseudorandomized conditions of difficulty in the experiment: 3, 4, 5, and 6 digits in the encoding set. Each condition consisted of 30 trials. Before the start of the experiment, participants completed a short training block with feedback on the accuracy of responses. Responses (correct/error) and reaction time, calculated as the time taken from the appearance of the probe to the button response, were recorded for all trials for each difficulty level.

Participants completed a Russian adaptation of the Mental Toughness Questionnaire (MTQ48; Clough et al., 2002), a 48-item inventory that requires responses to statements on a 5-point Likert scale ranging from strongly disagree, to strongly agree. The MTQ48 provides a total mental toughness (MT) score and measures six subscales of challenge, commitment, emotional control, life control, confidence in abilities and interpersonal confidence. The factor structure, validity and; reliability of the MTQ48 questionnaire were supported by previous studies [64,69].

The Raven’s Standard Progressive Matrices (RSPM) measure general intelligence as a basic cognitive function. Raven’s Matrices is a visual-spatial test that includes five sets of twelve different matrices of increasing in difficulty [70] and is designed to measure eductive (i.e., the ability to make meaning out of confusion) and reproductive (i.e., the ability to absorb, recall and reproduce information) abilities [71]. RSPM consists of 60 non-verbal logical-mathematical problems divided into five sets of 12 problems each (sets A, B, C, D and E), thereby increasing the difficulty both within and across sets [72]. Each problem includes a matrix of geometric figures with one pattern missing. The participant is asked to select the correct missing pattern of each matrix from a set of examples [73]. To obtain the final score, the total amount of points received is calculated (highest score—60 points).

### 2.3. EEG Equipment

EEG data were recorded from 64 electrodes placed according to the international 10-10 system with a Brain Products ActiChamp amplifier (BrainProducts, Munich, Germany). All experiments were performed in a soundproofed and electrically shielded room with dim lighting. The raw signal was recorded without any filtering and the sampling rate was 1000 Hz. Electrode-to-skin impedance was kept below 10 kΩ for all channels with a highly conductive chloride gel.

### 2.4. Data Analysis

EEG preprocessing was performed with Brainstorm software and Python scripts (including the MNE-Python package, [74]). Based on visual analysis, high-amplitude artifacts were rejected from the data. Prior to the Independent Component Analysis (ICA) analysis, bad channels were removed from EEG. ICA ocular correction analysis was then performed, and components related to eye movements were manually selected and removed from the data. After ICA, signals in bad channels (e.g., the signal looks flat) were replaced by spherical interpolations over the neighborhood electrodes. After removing artifacts, the following number of trials remained in the data: mean = 25 ± 1 for the 3-digit condition, mean = 34 ± 2 for the 4-digit condition, mean = 28 ± 2 for the 5-digit condition, and mean = 32 ± 3 for the 6-digit condition. In order to reduce volume conduction effects, current source density (CSD) transformation was applied to EEG data. Stimulus-locked epochs for each task difficulty condition were extracted from the data (−300 to 5000 ms relative to the stimulus). Trials with response times exceeding 3 sigma deviations from the mean were excluded from the analysis.

CSD signal in each channel was translated into the time-frequency domain using wavelet transformation with a multitaper approach (tfr_multitaper function from the MNE-Python package). The 7000 ms were used with a window centered at each 1000 ms time step from 1 to 40 Hz. A set of good tapers (i.e., those with the least leakage from far away frequencies) were chosen automatically based on MNE algorithms. The width of wavelets was different for different bands and was chosen so that the number of cycles was equal to frequency/2, where the frequency was a range from 1 to 40 Hz. For each time-frequency bin and each electrode, spectral power averaged over subsets of trials used for the analysis was calculated.

Finally, baseline normalization of power was performed, thus obtaining event-related spectral perturbation. To calculate ERD/ERS, the log-ratio baseline was calculated as an averaged spectral power over the −200 to 0 ms pre-stimulus time window (separately for each location and each frequency) and applied to the spectral power values. Statistical analysis was conducted using electrodes selected a priori and confirmed by the peak ERD/ERS of the alpha rhythm related to the baseline on the topography figures (Figure 2). A priori, alpha rhythm depression was expected on the parietal lateral electrodes (P3, P4), whereas alpha rhythm activation was expected to be observed on the parietal-occipital lateral electrodes (PO7, PO8), in accordance with previous EEG studies using the Sternberg task [47,50].

For statistical analysis, the individual alpha frequency was used (individual peak of oscillations power in the frequency range from 8 to 13 Hz). Alpha power related to the baseline was averaged over the selected electrodes in the time range selected using time-frequency figures and cluster-based permutation *t*-tests (1000 permutations) using permutation_cluster_1samp_test function from the mne Python package, that identifies statistically significant clusters of activity associated with the event. The following time intervals of interest were chosen: 300–1300 ms for ERD during encoding, 2300–3300 ms for ERS during retention, 3900–4400 ms for ERD related to the probe for responding (see Figure 2).

For each task difficulty level for each participant, average accuracy calculated as the proportion of correct responses, reaction time, and alpha power (in electrodes and time periods of interest) were calculated using R studio. Outliers were removed using the 3 sigma rule for all analyses. The following values were removed as outliers for each variable: 1 value for the encoding-related ERD variable, 0 values for the retention-related ERS variable, 1 value for the probe-related ERD variable, 9 values for the reaction time variable, 7 values for the accuracy variable, 4 values for the Raven score variable, 4 values for the mental toughness score variable.

Below we describe the analysis used to test each hypothesis:ERD during encoding, ERS during retention, and ERD related to responding are related to task accuracy.
A linear mixed effects model was fitted to test the effect of each EEG variable (e.g., ERD related to encoding) as a function of accuracy, using the following formula: a target variable—the EEG component, a predictor—accuracy (correct or error response), a random effect—participant ID. This random effect was used to control for individual differences in the EEG effect that can increase individual variability. The Wald z-statistic was used to test the coefficients of the model.For all subsequent analyses, only correct responses were selected.
2.ERD during encoding, ERS during retention and ERD related to responding increase with Sternberg task difficulty.
For each EEG variable, a linear mixed effects model was fitted (using nlme package in R) with the following formula: a target variable—the EEG effect calculated for each participant, a predictor—task difficulty, a random effect—participant ID. To test the coefficients of the model, the Wald z-statistics was used.
3.For the easier task conditions, the variability of the severity of these effects is explained by the Raven scores
For each EEG variable, a linear mixed effects model was estimated with the following formula: a target variable—the EEG scores, a predictors—task difficulty, Raven score (high or low) and their interaction, a random effect—participant ID. For these analyses, participants were divided into high-Raven (top 30%) and low-Raven (bottom 30%) groups using their total Raven score. To test the coefficients of the model, the Wald z-statistics was used.
4.For more difficult task conditions, the variability of the severity of these effects is explained by the mental toughness (MT) scores
For each EEG variable, a linear mixed effects model was with the following formula: a target variable—the EEG effect, predictors—task difficulty, mental toughness score (high or low) and their interaction, a random effect—participant ID. For these analyses, participants were divided into high-MT (top 30%) and low-MT (bottom 30%) groups using their total MTQ-48 score. To test the coefficients of the model, the Wald z-statistics was used.

For each fixed effect of each model, the effect size was calculated using the following formula: f^2^ = (R_2_^2^ − R_1_^2^)/(1 − R_2_^2^), where R_2_^2^ represents the variance explained for a model with the given effect and R_1_^2^ represents the variance explained for a model without the given effect [75]. This measure reflects the proportion of variance explained by the given effect relative to the proportion of outcome variance unexplained and is considered small at a value of 0.02, medium at a value of 0.15, and large at a value of 0.35 [76].

## 3. Results

### 3.1. Behavioral Basic Statistics

Mean accuracy for the 3-digit, 4-digit, 5-digit and 6-digit conditions was 93.5 ± 4.1%, 95.1 ± 4.5%, 95.1% ± 4.5%, and 94.4% ± 5.4%, respectively (Figure 3). The mean reaction time for the 3-digit, 4-digit, 5-digit and 6-digit conditions was 716 ± 146 ms, 773 ± 158 ms, 791 ± 162 ms, and 838 ± 179 ms, respectively (Figure 3).

The mean total MTQ-48 score of participants was 31.7 ± 4.2. The mean Raven score was 49.3 ± 6.5. Significant correlations were not observed among total scores on MTQ-48, Raven and Sternberg task accuracy when considering average accuracy and accuracy calculated for all task difficulty levels separately). Both mental toughness (r = −0.2165, *p* < 0.001) and Raven scores (r = −0.2375375, *p* < 0.001) showed significant negative relations with reaction time on the Sternberg task.

### 3.2. Alpha Power Correlates of the Sternberg Task

During the encoding period (0–1.5 s after stimulus onset), alpha power ERD was observed in the 300–1300 ms time interval with a peak on the parietal lateral electrodes (Figure 3). During the retention period (1.5–3.5 s), alpha power ERS was observed in the 2300–3300 ms time interval with a peak on the occipital-parietal lateral electrodes (Figure 3). During the probe period (3.5–5.5 s), alpha power ERD was observed in the 3900–4400 ms time interval with a peak on the parietal lateral electrodes (Figure 2).

### 3.3. Accuracy Dependence on Alpha Power Correlates of the Sternberg Task

Linear mixed effects model predicting the ERD related to encoding showed a significant reduction in the effect for erroneous responses, in comparison with correct ones (t = 2.298, *p* < 0.05, f^2^ = 0.01). Linear mixed effects model predicting the ERS related to retention did not reveal significant differences between erroneous and correct responses (t = −1.261, *p* > 0.05, f^2^ = 0.003). Linear mixed effects model predicting the ERD related to a probe showed significant reduction in the effect for erroneous trials (t = 2.619, *p* < 0.01, f^2^ = 0.01; Figure 4).

### 3.4. Alpha Power Correlates Dependence on Task Difficulty

Linear mixed effects models revealed a significant negative relation between task difficulty and ERD related to encoding (t = −8.497, *p* < 0.001, f^2^ = 0.25; Figure 5A). A significant positive relation was observed between task difficulty and ERS related to retention (t = 2.806, *p* < 0.005, f^2^ = 0.03; Figure 5A). Significant negative relation was revealed between task difficulty and ERD related to a probe (t = −2.103, *p* < 0.05, f^2^ = 0.02; Figure 5A).

### 3.5. Raven Scores Influence on Relation between EEG Effects and Task Difficulty

Significant negative relation was revealed between the Raven score (high/low Raven groups condition) and the event-related alpha power during encoding (t = −2.02, *p* < 0.05, f^2^ = 0.02; Figure 5B). No significant effect was found for Raven scores and the task difficulty predictors interaction (t = 0.725; *p* > 0.05, f^2^ = 0.01; Figure 5B).

No significant relation was found between ERS during retention and the Raven score (t = −1.557, *p* > 0.05, f^2^ = 0.02). No significant effect was found for Raven scores and task difficulty predictors interaction (t = 0.508, *p* > 0.05, f^2^ = −0.004). No significant relation was observed between ERD during a probe and Raven scores (t = −1.101, *p* > 0.05, f^2^ = 0.02). No significant effect was found for Raven scores and the task difficulty predictors interaction (t = 0.637, *p* > 0.05, f^2^ = −0.003).

### 3.6. Mental Toughness Scores Influence on Relation between EEG Effects and Task Difficulty

In the probe-related alpha ERD analysis, a significant interaction effect was found for mental toughness scores and the task difficulty predictors: low mental toughness increased the negative relation between alpha power and task difficulty (t = −3.282, *p* < 0.005, f^2^ = 0.06; Figure 5B), in other words, low mental toughness participants, as compared to high mental toughness ones, demonstrated ERD in difficult task conditions.

No significant relation was found between the ERD related to encoding and the mental toughness score (high/low MTQ48 groups condition; t = −0.707, *p* > 0.05, f^2^ = 0.02). No significant effect was found for the mental toughness score and the task difficulty predictors interaction (t = 1.736, *p* = 0.08, f^2^ = 0.005). No significant relation was found between the ERD related to retention and the mental toughness score (high/low MTQ48 groups condition; t = −1.435, *p* > 0.05, f^2^ = 0.02). No significant effect was found for the mental toughness score and the task difficulty predictors interaction (t = 0.696, *p* > 0.05, f^2^ = −0.006).

No significant relation was found between the ERD related to a probe the mental toughness score (high/low MTQ48 groups condition; t = −0.338, *p* > 0.05, f^2^ = 0.02).

## 4. Discussion

We investigated parietal alpha oscillations and their relations to task difficulty as measured using the Sternberg task, Raven scores and mental toughness. We highlight five main findings. (a) As expected we found reaction time increased and accuracy decreased with task difficulty, with the exception of the easiest level that had lower accuracy compared other levels; (b) alpha power correlates demonstrate classic oscillatory effects for encoding (early ERD), retention (ERS) and probe (late ERD), replicating past research; (c) task difficulty modulated oscillatory effects, albeit only for the ERD related to encoding this correlation was strong and linear; (d) participants with high Raven scores demonstrated significantly stronger ERD related to encoding; € participants with high mental toughness scores demonstrated significantly stronger ERD related to responding at for most difficult task conditions. Findings contribute to our understanding of cognitive mechanisms and neuromodulation and are discussed in terms of theories of effort.

### 4.1. Behavioral Analysis

Participants’ mean accuracy scores on the Sternberg task were high (>90%) for all difficulty levels, confirming that participants understood the task. The reaction time cost was about 20–50 ms as latencies decreased with difficulty. Although the reaction time to the Sternberg task showed a linear increase as a function of difficulty, participants had significantly lower accuracy in the easiest condition (3-digits). Probably participants erroneously underestimated this condition as too easy and failed to allocate adequate control during performance. It is known that people find very easy tasks to be boring, monotonous and even unpleasant, which may lead participants to avoid the task [77]. This may explain the decreased accuracy in this, the easiest condition.

As expected, mental toughness did not correlate with the Raven scores. However, mental toughness and Raven scores were negatively related to reaction time, but not accuracy. A lively debate exists on whether reaction time can be interpreted as effort, since long reaction times may reflect both increased cognitive load (increased effort) and attentional failure (insufficient effort level; [62,78]). Nevertheless, our results suggest that both mental toughness and nonverbal intelligence are independently important for Sternberg task performance. According to chronometric approaches to human intelligence [79,80], intelligence is directly related to the speed of information processing. This assumption has been supported by many studies, which led to the view that the shorter reaction time in participants with high intelligence levels could correspond to increased efficiency of brain functioning [56].

### 4.2. Alpha Power Correlates of the Sternberg Task

We observed classic oscillatory correlates of the Sternberg task condition in the alpha band: parietal ERD related to encoding, parietal-occipital ERS related to retention, and parietal ERD related to responding to a probe. These results concur with findings in magnetoencephalography (MEG) and EEG studies [8,43,47,49,50]. ERD related to encoding and retrieval may be interpreted as cortical activation [10] and visual attention enhancement required during information encoding and/or information retrieval in working memory [10,44,81,82,83]. ERS related to retention may be interpreted as cortical deactivation and inhibitory control required to inhibit irrelevant information so as to keep the relevant information in working memory [10,84,85]. It should also be noticed that ERS was pronounced in a slightly different alpha frequency band—upper alpha frequency (10–13 Hz), in comparison with ERD. This effect was observed also by Wianda and Ross [8], who concluded that ERS related to retention effect reflects a separate functional mechanism of alpha oscillations being not simply opposite to ERD related to encoding.

### 4.3. Accuracy Dependence on Alpha Power Correlates of the Sternberg Task

Only ERD effects showed significant relation to accuracy. Weak ERD increased the likelihood of an error. This suggests that ERD related to set encoding and responding to the probe is critical for task performance. We also observed a similar trend for the ERS related to retention: weak ERS seemed to insignificantly increase the probability of an error. Our results suggest that successful performance on the Sternberg task relies more on correct information encoding and retrieving in working memory than information retention. This result may be explained by the features of our task requiring participants to retain information for only two seconds with no distractors. An alternative explanation may be that the ERD effect related to encoding overlapped the ERS effect since the ERD preceded ERS and affected electrodes used for ERS analysis (PO7–PO8). The relative duration of the retention phase may lead to better dissociation of these alpha correlates of encoding and retention. However, behavioral statistics show a low percentage of errors for all task conditions, which probably led to increased variance in the error responses group, thus results need to be replicated. To our knowledge, there are no studies that have shown an association between accuracy and the power of the parietal alpha rhythm associated with working memory task performance. Nevertheless, the encoding and retention processes reflected by ERD and ERS are considered necessary to perform the Sternberg task [8,43,47,49].

### 4.4. Alpha Power Correlates Dependence on Task Difficulty

ERD related to encoding increased strongly and linearly with task difficulty, which is in agreement with previous studies [35]. This EEG effect may reflect a resource-intensive top-down process whose activation level depends on task requirements. This result may be interpreted by existing theories of effort as the investment of resources that enable the execution of behavior, arguing that effort functions to sustain activity that is needed for goal attainment [12,63]. According to existing theories of mental effort [62], the resources needed to activate the top-down control processes are limited, which requires their economy and rational use. Therefore, instead of always performing a task at the maximum level of their capabilities, a person increases the efforts invested in completing a task as its requirements increase.

ERD related to the probe was significantly increased with task difficulty as well. The relation between task difficulty and ERD during response preparation and execution in the Sternberg task has not been shown to date. Parietal ERD associated with a recognition and response probe appears to reflect the evidence accumulation process during decision-making [86]. This relation suggests a correspondence of effort modulation to the mobilization of resources. However, the enhanced ERD related to the probe was observed only for the most difficult condition. Thus, the classic idea about effort as a general phenomenon may be incorrect: different processes required by a task may be differently related to effort. Perhaps the processes reflected by ERD related to the probe (attentional preparation, information retrieving) are less effortful, which is why we did not observe an effort increase at easy conditions (3, 4, 5 digits) for which the base ERD level was enough. However, weak enhancement of ERD may be also explained by individual differences in the mental toughness of participants. Specifically, ERD and difficulty correlated for participants with low mental toughness, but not for the high mental toughness group.

ERS related to retention increased with task difficulty, which is in agreement with previous studies [46,47]. We can interpret this relation in terms of the increased effort needed to mobilize cognitive resources. However, the ERS level did not differ between the two most difficult task levels (5 and 6 digits). This result may be explained by the ceiling effect: on average, participants were not capable to activate ERS more. It would mean that the process reflected by ERS (inhibition, relevant information retention) requires too many cognitive resources so that even 6 digits could not have been maintained at the proper level by activation in this posterior brain region. This is consistent with functional magnetic resonance imaging results showing a leveling off of activation in the parietal cortex when five or six items need to be processed, whereas activation in prefrontal regions continued to increase linearly [87]. Future research should consider brain-behavioral relations in prefrontal areas. Another explanation may be that we observed the phenomenon of effort drop described by the Motivational Intensity Theory [88]. According to this theory, individuals stop investing effort in task performance, when the task is too difficult, and the likelihood of success is assessed as too low. Perhaps, the processes reflected by ERS are too resource-intensive and the 6-digit level of the Sternberg task was enough to stop investing in these processes. However, this is unlikely because the performance in the highest difficulty level was higher than 90%. Overall, both interpretations suggest a high resource cost of the retention process implemented through inhibitory control. It is important to note that our results suggest that while participants invest in some particular processes, they may stop investing in other ones, thus, effort may not be a general phenomenon.

Critically, the alpha ERD effect that shows increases with task difficulty is also observed in studies using other working memory tasks, for example, the n-back task [32,34,40,41,46]. However, studies using the n-back task do not report results associated with an alpha ERS effect. Although the Sternberg task is similar to the n-back task in requiring encoding, retention, and retrieving, the n-back task requires simultaneous implication of these processes, which leads to overlapping in some of their neural correlates [46]. Therefore, the increasing alpha ERD with task difficulty can be associated with both encoding and retrieving and can also be distorted by the overlapping ERS effect. Interestingly, Fairclough and Ewing [41] observed an increase in absolute alpha power in the most difficult 7-back condition. This effect may indicate the drop in effort in the very difficult condition, as described by the Motivational Intensity theory. We suggest that a study using the Sternberg task that includes very difficult conditions and ERD/ERS measures may more clearly reveal this effect as a stimulus-related correlate of task performance and show which specific process this effect refers to.

### 4.5. Mental Toughness Scores Influence on Relation between EEG Effects and Task Difficulty

We also hypothesized that if alpha correlates of the task reflect the effort, we would observe the influence of mental toughness on these effects for the most challenging task levels. Mental toughness reflects both internal motivation to perform challenging tasks and self-efficacy [5,64], thus, we assumed that at the most difficult levels, participants with mental toughness would demonstrate stronger alpha effects due to stronger effort.

Indeed, we observed a significant influence of the mental toughness scores in the most difficult conditions. However, the effect was contrary to our hypothesis: participants with lower scores on mental toughness demonstrated stronger ERD related to the probe, which may be interpreted as stronger effort. Perhaps, our task was too easy to reveal the effect predicted by our hypothesis based on the literature review [12,56,63]: increased effort for more motivated and emotionally stable participants at challenging conditions. Alternatively, due to their low self-efficacy, individuals with low mental toughness assessed the 5- and 6-digit conditions as difficult, but feasible and, thus, requiring effort increase, while individuals with high mental toughness levels assessed these conditions as easy and not requiring effort increase. Then it can be hypothesized that increasing the encoding set and adding more difficult conditions would reveal an expected phenomenon of effort increase for people with high mental toughness levels and effort drop for people with low mental toughness. This hypothesis concerns only effort investment into processes involved in the last step of the Sternberg task: attentional preparation, and response execution.

We did not observe significant differences in the ERD related to encoding and the ERS related to retention. Thus, ERD related to responding to the probe is getting enhanced only for the high-mental toughness group, whereas ERD related to encoding is getting enhanced for everyone. Perhaps these emotional-motivational aspects affecting effort appear only at the latest stage of the task when a response is required. The fact that we do not see mental toughness effect on the ERS, may mean that the ERS dynamic we observed (ERS stops increasing at the most difficult conditions) cannot be explained by effort leveling off, and rather should be explained by the ceiling effect.

### 4.6. Raven Scores Influence on Relation between EEG Effects and Task Difficulty

We evaluated intelligence to control for its effects on task performance in relation to mental toughness. Previous studies have shown differences in the alpha rhythm between participants with high and low levels of intelligence (the Raven test was mainly used; [57,89]. However, for subjectively difficult tasks, these differences might be explained by different motivational levels in the compared groups [56].

According to our hypothesis, if the EEG effect reflects effort, it should be different between participants who score higher or lower on The Raven’s Standard Progressive Matrices as an intelligence measure. According to the neural efficiency effect, individuals with higher intelligence invest less effort in task performance, in general, since these tasks are easier for them [11]. However, only the ERD related to encoding showed a significant difference in values between the high- and low-Raven groups. The low-Raven group demonstrated weaker ERD, which was opposite to our expectations. The decrease in ERD related to encoding in low-Raven participants contradicts the hypothesis of Neubauer and Fink [11] that the neural efficiency effect may be observed for easy tasks. Perhaps, this type of brain activation was assessed by the low-Raven group as too resource-intensive, so they tried to avoid it and use different strategies.

We did not observe significant differences between high- and low-Raven groups for ERS related to retention and ERD related to a probe. It means that the processes reflected by these EEG-correlates were activated at the same level for these two groups. It suggests that the capability to activate these processes (top-down inhibition and attentional preparation and information retrieval) does not depend on figurative intelligence. Another explanation may be that the difference in these effects was not observed, because this difference occurs in the first trial phase: participants with high intelligence invested strongly in the first required process (encoding) and did not need to invest strongly in the subsequent phases of a trial (retention and the probe).

An important result for our study is that the Raven influence on the studied alpha correlates differed from the mental toughness influence, which may mean successful dissociation of the cognitive and motivational factors affecting performance.

### 4.7. Effort Distribution among Processes

Many studies consider effort as a general phenomenon related to task performance and study it through general physiological correlates of task performance, such as cardiovascular indicators or general cortical activation [40,41,90,91]. However, some authors suggest that not all cognitive processes require effort, while some may require more effort than others, and the performance of any task is associated with the distribution of effort among the controlled processes [62]. Thus, it would be more correct to study effort through physiological correlates of separate cognitive processes required by a task. Different results for the studied EEG effects of encoding, retention and recognition may mean different relations among these processes and effort. It may mean that effort is distributed differently for different working memory processes.

For encoding, effort increases with objective task difficulty. For retention (i.e., maintenance), this increase in effort levels at the level of 6 digits in parietal cortices is probably due to the high cost of this process. Thus, this result leads to two hypotheses: (1) retention is a more resource-intensive process than encoding, and (2) distribution of effort depends on the resource intensity of a process.

According to our results, for processes related to attentional preparation and information retrieval, the increase in effort was pronounced only in people with low mental toughness. Thus, mental toughness determines the effort modulation dynamics. The mechanism of effort distribution in parietal cortices is different for different processes: for some processes, mental toughness plays a role, while in others it does not. It also means that the Motivational Intensity Theory [12] does not account for some psychological differences related to difficulty perception and leads to a stronger increase in effort with task difficulty in people who are less resistant to difficulties.

### 4.8. Limitations

We recognize three limitations of our study. First, the experimental task albeit having 4 levels of difficulty appeared to be too easy for participants and led to a low number of errors, which made it difficult to interpret the relations between the alpha rhythm and accuracy. Second, the absence of more difficult conditions did not allow us to interpret the relation of mental toughness with alpha rhythms in terms of the Motivational Intensity Theory, which would be highly relevant in this effort study. Third, short time intervals related to the retention phase made it difficult to interpret the functional significance of alpha ERS. As this is the first study to identify relations between alpha power and personality traits such as mental toughness, future research is needed to replicate and expand these results. Future studies should consider including reward manipulations and additional questionnaires that distinguish intrinsic and extrinsic motivation.

## 5. Conclusions

We investigated how ERD related to encoding, ERS related to retention, and ERD related to recognition of a probe depend on task difficulty, mental toughness, and cognitive ability. All studied alpha effects increased with task difficulty and are considered as correlates of the top-down resource-intensive processes necessary for task performance, illustrating dynamics as modulation of effort. However, for ERD related to a probe, this relation was observed only in participants with low mental toughness, showing that the modulation of effort depends on the motivational-emotional response to task difficulty: participants who are less resistant to challenging tasks paradoxically invest more in rather difficult tasks. This result contradicts the predictions of Motivational Intensity Theory. For the further study of this phenomenon, more difficult task conditions should be added to the experimental design. Our results show that the relation between task difficulty, intelligence, mental toughness and stimulus-related alpha power varies with the type of cognitive process (encoding, retention and recognition). This suggests that the effort distribution mechanism works differently for different cognitive processes. Therefore, it can be concluded that effort should not be viewed as a general process associated with task performance and must be studied through correlates of separate processes. Further study of factors that drive effort modulation requires new experiments with more challenging conditions, alternative measures of intrinsic motivation, and manipulation of extrinsic motivation. Such investigations can reveal the influence of various motivational factors on effort in particularly difficult conditions.

## Figures and Tables

**Figure 1 brainsci-12-01135-f001:**
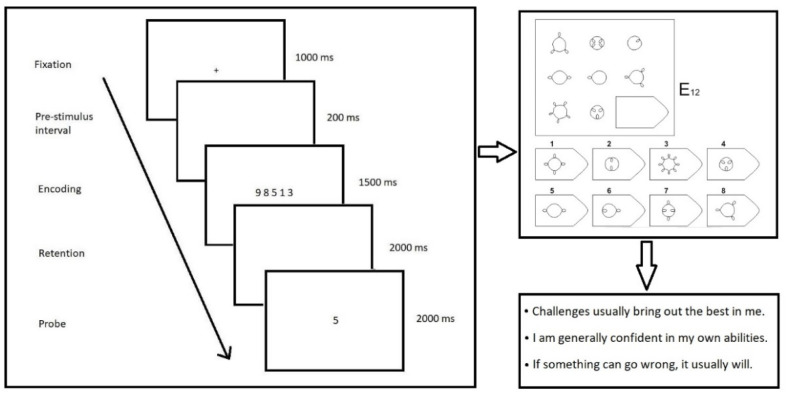
Research materials. (**Left panel**): Example of stimuli and presentation sequence for the Sternberg task. (**Right panels**): (**top**)—example of the RSPM stimuli; (**bottom**)—example of the MTQ48 questions.

**Figure 2 brainsci-12-01135-f002:**
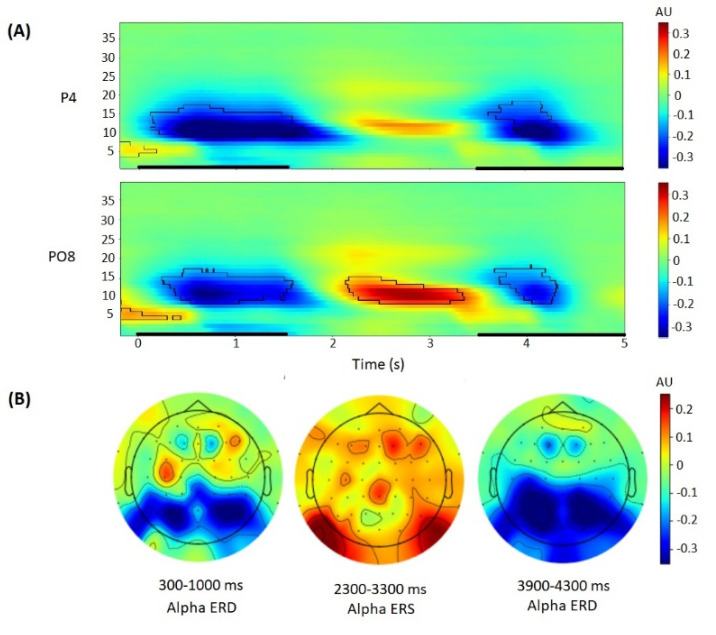
EEG oscillatory activity during Sternberg task trial performance. (**A**): Time-frequency plots of baseline-corrected oscillatory activity at the P4 (**top**) and PO8 (**bottom**) electrodes. Black outlines indicate *p* < 0.05 (permutation statistics). Time is shown relative to the stimulus onset. Black bold lines on *x*-axis indicate time periods of stimulus presentation for the encoding and probe stages. (**B**): Topographical maps of baseline-corrected alpha band activity at 300–1000 ms, 2300–3300 ms, 3900–4300 ms after stimulus onset.

**Figure 3 brainsci-12-01135-f003:**
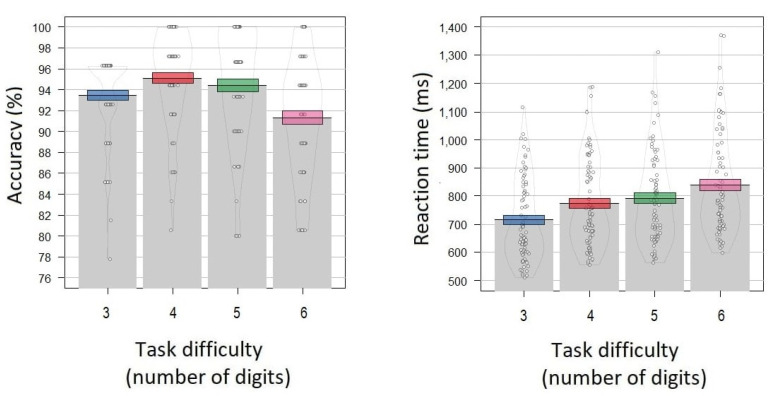
Behavioral results of the Sternberg task. (**Left panel**): *y*-axis—accuracy (percentage of correct responses averaged across participants). (**Right panel**): *y*-axis—reaction time averaged across participants. *x*-axis—the task difficulty condition (3, 4, 5, 6 digits in the encoding set). Grey bars and black lines indicate means. Colored bands indicate ±1 standard errors (blue color corresponds to the 3 digits condition, red—4 digits, green—5 digits, pink—6 digits). Dots and density curves indicate data distribution.

**Figure 4 brainsci-12-01135-f004:**
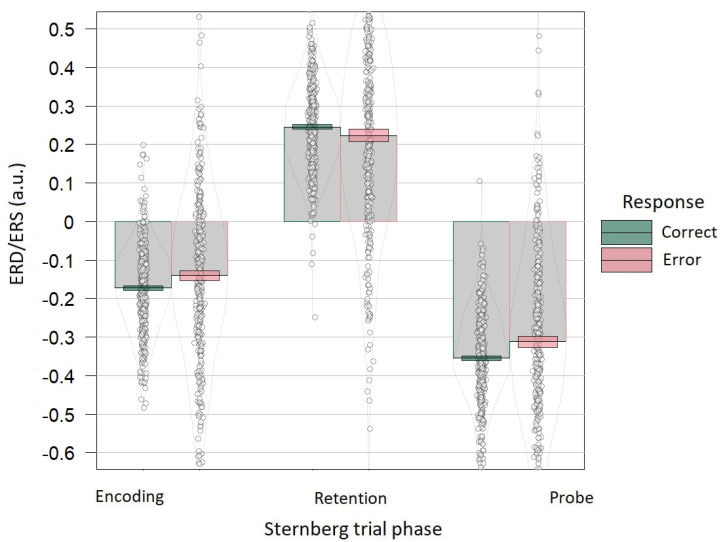
Event-related alpha power by response accuracy shown separately for each Sternberg trial phase. During the encoding phase, ERD was observed (300–1300 ms), during the retention phase ERS was observed (2300–3300 ms), during the probe phase, ERD was observed (3900–4300 ms). For this visualization, for each trial phase, the mean of the alpha power was normalized to minimize individual differences and added to the initial mean ± 1 standard error of the mean. Grey bars and black lines indicate means. Colored bands indicate ±1 standard errors. Dots and density curves indicate data distribution.

**Figure 5 brainsci-12-01135-f005:**
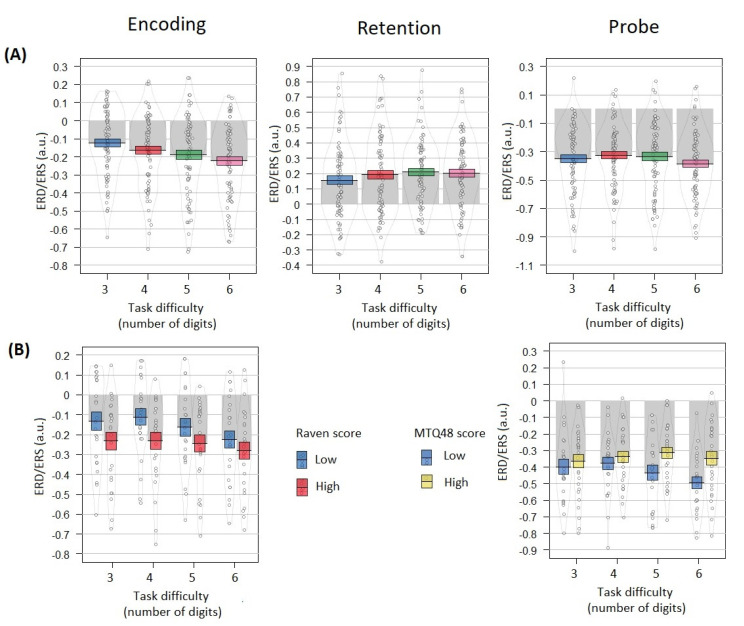
Event-related alpha power by task difficulty and individual differences shown separately for each Sternberg trial phase: left vertical panel—Encoding (300–1000 ms), middle vertical panel—Retention (2300–3300 ms), right vertical panel—Probe (3900–4300 ms). (**A**) Relation of alpha ERD/ERS with task difficulty. Colored bands indicate ±1 standard errors (blue color corresponds to the 3 digits condition, red—4 digits, green—5 digits, pink—6 digits). (**B**) Interaction between task difficulty and individual differences groups (‘High’—top 30% Raven/MTQ48 score group, ‘Low’—bottom 30% Raven/MTQ48 score group). Only significant results are shown: for the encoding phase, significant interaction between difficulty and Raven score groups was found; for the probe phase, significant interaction between difficulty and MTQ48 score groups was found. Grey bars and black lines indicate means. Dots and density curves indicate data distribution.

## Data Availability

Anonymous EEG and behavioral data can be provided by the authors of the article upon request.

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
