# Peer review of "Parietal Alpha Oscillations: Cognitive Load and Mental Toughness"

_brainsci, 2022, doi:10.3390/brainsci12091135_

Round 1

Reviewer 1 Report

1.      The uppercase and lowercase of the title should be revised based on MDPI format.

2.      Qualitative results need to be added in the abstract section.

3.      Keywords need to be reordered based on alphabetical order.

4.      Please remove numbers after keywords and use the lowercase font, such as “Effort 1” to “effort”.

5.      The present work is lack of novel. Studies related to parietal alpha have been widely studied in the past. Nothing really new exists in the present study. The authors mandatory need to highlight their novelty in the introduction section.

6.      The authors need to explain the previous study with their findings and its limitation to show up the research gaps. For example, A studied…. Their findings…….. Study limitation………. The present introduction does not explain it.

7.      In lines 53-56, Where is cognitive load? The authors have mentioned mental toughness as the present title but not with cognitive load.

8.      Regarding mental toughness, autism spectrum disorder is one of the issues related that should be mentioned in the introduction and/or discussion section. The authors need to deliver these important points. Also, to support this explanation, additional reference published by MDPI needs to be adopted as follows: Physiological Effect of Deep Pressure in Reducing Anxiety of Children with ASD during Traveling: A Public Transportation Setting. Bioengineering 2022, 9, 157. https://doi.org/10.3390/bioengineering9040157. This reference also would be adopted in line 38 for explaining task performance since the authors used reference [1] twice in line 37 too.

9.      The introduction section needs to be shortened and make it more concise. Cut off not necessary things. Not using subsection is really recommended.

10.   The research flow needs to be explained as a form of illustration figure in the materials and methods section.

11.   In subsection 2.1 participants, what is the basis for using the number of the participant? What is the basis, standard, and regulation?

12.   Related to previous comments, the authors used a very small sample which would be leading to biased results since the sample used was very impactful in the study.

13.   The limitation of the present research should be explicitly explained before the conclusion section.

14.   Further research needs to be mentioned in the conclusion section.

15.   English proofreading is mandatory to solve grammatical errors and the language style used.

16.   Please used MDPI template properly, the present form does not. For example many errors in typesetting.

Author Response

Thank you for your constructive comments and suggestions for improving our article. We have addressed all comment and made corresponding edits in the text as detailed below. Please note that the line numbers correspond to the tracked version of the text. 

  1. The uppercase and lowercase of the title should be revised based on MDPI format.

Response: Thank you. We have revised the title according to MDPI format. “Parietal Alpha Oscillations: Cognitive Load and Mental Toughness”

  1. Qualitative results need to be added in the abstract section.

Response: Thank you for the comment. We have added quantitative results in the abstract in lines 25-27. “A positive dependence on difficulty was observed for all studied oscillatory effects (t=-8.497, p<0.001; t=2.806, p<0.005; t=-2.103, p<0.05). The influence of Raven intelligence was observed for encoding-related alpha ERD (t=-2.02, p=0.049). The influence of MT was observed only for difficult conditions in recognition-related alpha ERD (t=-3.282, p<0.005). “

  1. Keywords need to be reordered based on alphabetical order.

Response: Thank you. We have reordered keywords in alphabetical order. Lines 30-31. “Keywords: alpha oscillations; effort; encoding; mental toughness; recognition; retention; task difficulty; working memory “

  1. Please remove numbers after keywords and use the lowercase font, such as “Effort 1” to “effort”.

Response: Thank you. We have removed numbers after keywords and used lowercase font. Lines 32-34. “Keywords: alpha oscillations; effort; encoding; mental toughness; recognition; retention; task difficulty; working memory “

  1. The present work is lack of novel. Studies related to parietal alpha have been widely studied in the past. Nothing really new exists in the present study. The authors mandatory need to highlight their novelty in the introduction section.

Response: Thank you for the comment. We take this opportunity to highlight that this is the first study that examines the impact of mental toughness and intelligence on the relation between alpha power and task difficulty. Further, we recognize that many studies examine parietal alpha reflecting working memory processes and some of them use the Sternberg task, however the knowledge on process-specific factors and parametric modulations of difficulty is still limited. Specifically, some of studies investigate the relation between cognitive load and alpha event-related (de)synchronization (ERD/ERS) associated with encoding, retention, and recognition. Although studies have separately examined ERD related to encoding and ERS related to retention increases with task difficulty, studies that examined both these effects on the same sample did not reveal significant relation between encoding-related ERD and task difficulty. To our knowledge our study is the first to reveal both these effects related to the Sternberg task performance. The lack of such results in the literature may be due to past studies using few difficulty levels in their experimental design (i.e., usually two levels, rarely three levels). Thus, little is known about parametric changes in parietal alpha as a correlate of cognitive load. Our study examines for the first time effects of ERD/ERS across four levels of difficulty and across three cognitive processes (encoding, retention and responding). Understanding relations of parietal alpha with cognitive load, mental toughness and intelligence can inform theories of cognition and motivation. Practically our results can benefit future research on cognitive effort and neuromodulation. 

We have improved the narrative to highlight the novelty of our study.

Lines 132-144: “ERD related to encoding [57] and retrieving [55] and ERS related to retention [18, 55, 56] increases with task difficulty. Critically, no study has revealed in the same sample both effects when examining Sternberg task performance. Specifically, studies that examined both these effects did not reveal significant relations between encoding-related ERD and task difficulty [8, 18, 20, 55, 56]. Further, most studies do not calculate individual alpha frequency for their analyses, while alpha frequency peak is known to show large individual differences and, thus, analysis of fixed frequency ranges may not be useful for studying alpha power modulation effects [9].  Our study examines for the first time effects of ERD/ERS across four levels of difficulty and across three cognitive processes (encoding, retention and responding). A main goal of our study is to identify how ERD/ERS related to difficulty and cognitive process are modulated by individual difference in mental toughness and intelligence.”;

Linea 212-215: “Importantly, according to our knowledge, no study has dissociated the impact of in-dividual differences associated with mental toughness and intelligence on the relation between alpha power and task difficulty.”

  1. The authors need to explain the previous study with their findings and its limitation to show up the research gaps. For example, A studied…. Their findings…….. Study limitation………. The present introduction does not explain it.

Response: Thank you for the constructive comment. We have revised information provided in the introduction to restructure the narrative in the format you have suggested. 

Lines 135-140: “Specifically, studies that examined both these effects did not reveal significant relations between encoding-related ERD and task difficulty [8, 18, 20, 55, 56]. Further, most studies do not calculate individual alpha frequency for their analyses, while alpha frequency peak is known to show large individual differences and, thus, analysis of fixed frequency ranges may not be useful for studying alpha power modulation effects [9] “;

Lines 184-188: “Many studies that examined the relation between alpha power and task difficulty [15, 60, 61, 63-65] did not control for motivational factors. Without considering personality traits associated with motivation, it is difficult to determine whether the modulation of alpha activity as an effort indicator is associated with emotional-motivational aspects or with merely cognitive capabilities of participants.”

  1. In lines 53-56, Where is cognitive load? The authors have mentioned mental toughness as the present title but not with cognitive load.

Response:  Thank you for the thoughtful comment. We have added the information about task difficulty factor in our research to this sentence (lines 55-58).

  1. Regarding mental toughness, autism spectrum disorder is one of the issues related that should be mentioned in the introduction and/or discussion section. The authors need to deliver these important points. Also, to support this explanation, additional reference published by MDPI needs to be adopted as follows: Physiological Effect of Deep Pressure in Reducing Anxiety of Children with ASD during Traveling: A Public Transportation Setting. Bioengineering 2022, 9, 157. https://doi.org/10.3390/bioengineering9040157. This reference also would be adopted in line 38 for explaining task performance since the authors used reference [1] twice in line 37 too.

Response:  Thank you. Although our study examined typical healthy adults, we recognize that our findings can be of clinical importance. We have added reference to Afif et al. in the line 40.

  1. The introduction section needs to be shortened and make it more concise. Cut off not necessary things. Not using subsection is really recommended.

Response: Thank you. We have carefully edited the introduction to shorten the text and removed subsection.   

  1.   The research flow needs to be explained as a form of illustration figure in the materials and methods section.

Response: Thank you for your comment. We have integrated figure 1 that illustrated the Sternberg task with materials we used in the study that include examples of stimuli for the Raven Progressive Matrices and the structure of the Mental Toughness questionnaire.

  1.   In subsection 2.1 participants, what is the basis for using the number of the participant? What is the basis, standard, and regulation?

Response: Thank you for the thoughtful comment.  In our study we are reporting data on 80 participants. Although there is no perfect number for sample size in EEG studies, the notion that more is better is typically relevant. For instance, from the studies we are referring to in the text, the majority of EEG studies that examine cognitive load use sample sizes that range between 10-20 participants. The EEG studies we cite had an average of 35.6 ± 34.5 participants (median = 20). For instance, studies that examine individual differences in alpha rhythm associated with performance on cognitive tasks report an average of 46.3± 11 participants. Past studies recognize that EEG studies usually underestimate the power (Button et al., 2013, doi:10.1038/nrn3475) therefore, in our study we are reporting data from 80 participants. We have edited the text to clarify this point on lines 229-231: “The sample size was determined taking into account the previous EEG studies described in the Introduction, as well as the known understated power effect in EEG studies [73].”

  1.   Related to previous comments, the authors used a very small sample which would be leading to biased results since the sample used was very impactful in the study.

Response: Thank you for the comment. We have collected data from 80 young adult participants. This number is well above the average number of participants that is typically used in studies that examine parietal alpha and performance on cognitive tasks. We perform analyses using the whole sample and part of the sample. Critically, a smaller sample size poses a higher likelihood of false negatives (i.e., type 2 errors), rather than false positives (type 1 errors). As we detect significant effects in our split group analyses, we are confident that the results are reliable. As this is the first study that investigates relations of alpha power and personality traits, we improved the narrative to indicate that future research should replicate the result (lines 686-688)

  1.   The limitation of the present research should be explicitly explained before the conclusion section.

Response: Thank you for pointing this out. We have added a limitation paragraph at the end of Discussion section. Specifically, we identify three key limitations, first the low number of errors, which makes it difficult to interpret the relations between alpha rhythm and accuracy. Second, the absence of more difficult conditions, which would allow us to better interpret the relation of mental stability with alpha rhythms in terms of the Motivational Intensity Theory. Third, short time intervals related to stimulus retention in the experimental design, which make it difficult to analyze the functional significance of alpha ERS. As this is the first study to identify relations among alpha power and personality traits, future research should replicate these finding and take these points into account. The text was edited accordingly on lines 679-689: “We recognize three limitations of our study. First, the experimental task albeit having 4 levels of difficulty appeared to be too easy for participants and led to the low number of errors, which made it difficult to interpret the relations between the alpha rhythm and accuracy. Second, the absence of more difficult conditions did not allow us to interpret the relation of mental toughness with alpha rhythms in terms of the Motivational Intensity Theory, which would be highly relevant in this effort study. Third, short time intervals related to the retention phase made it difficult to interpret the functional significance of alpha ERS. As this is the first study to identify relations among alpha power and personality traits, future research is needed to replicate and expand these results. Future studies should consider including reward manipulations and additional questionnaires that distinguish intrinsic and extrinsic motivation.”.

  1.   Further research needs to be mentioned in the conclusion section.

Response: Thank you for the comment. Indeed, we had mentioned in the conclusion section that future studies on this topic requires new experiments with more challenging conditions, and alternative measures of intrinsic motivation, and manipulation of extrinsic motivation. Such investigations can reveal the influence of various motivational factors on effort in particularly difficult conditions. We take the opportunity to improve the narrative accordingly on lines 707-710: “Further study of factors that drive effort modulation requires new experiments with more challenging conditions, and alternative measures of intrinsic motivation, and manipulation of extrinsic motivation. Such investigations can reveal the influence of various motivational factors on effort in particularly difficult conditions.”.

  1.   English proofreading is mandatory to solve grammatical errors and the language style used.

 Response: Thank you. We have carefully read the manuscript and edited to improve language and grammar.

  1.   Please used MDPI template properly, the present form does not. For example many errors in typesetting.

Response: Thank you for pointing it out. We found seven errors in typesetting and corrected them in the text.

Reviewer 2 Report

Zhozhikashvili et al. investigated the role of parietal alpha rhythms to account individual differences in intelligence and mental toughness. 80 subjects were tasked with a Stenberg task, Raven Standard Progressive Matrices test and a mental thoughness test (MTQ48). EEG recorded during the encoding, maintenance and probe phases was analysed and alpha power related to these individual variables. 

The paper is well-written and the topic is relevant but, in my opinion, the drawbacks are some missing information and control analyses.

Introduction: 

The authors nicely present some of the classical views of the functional role of alpha synchronization and desynchronization. However, in my opinion a complete picture of the current knowledge of this functional role is missing because some seminal works related to alpha are not presented in the Introduction. For example, the role of alpha related to the inhibition of task-irrevelant areas (Jensen et al, 2010), idling (Pfurtscheller and Lopes da Silva, 1999) or the amount of information retrieved (Hanslmayr et al., 2012, 2016; Griffiths et al., 2019). Also, the exact functional significance of alpha oscillations is still a matter of debate and in my opinion, this should be noted in the Introduction. 

The authors examined the alpha correlates (ERD and ERS) found with the Stenberg task but they didn't relate them to the functional meaning of these correlates. Here I miss a brief explanation of the modulation of the alpha band in function of the memory load, distractors, memory performance, cognitive effort... In this study I found especially important the memory load modulation.

Data analysis:

Line 255: Crucial parameters for time-frequency decomposition are missing: the width of the wavelets in number of cycles, frequency band of interest, times on which the analysis windows was centered. 

Line 263: Please, provide the final number of clean trials per condition (mean and SD) that were used in the subsequent analyses.

Line 283: Please, provide information about the amount of data discarded.

Line 292: This was already stated before.

Results:

The authors do an a priori selection of the channels of interest based on previous literature. However, the topoplot of Figure 3B related to the maintenance stage shows a bilateral alpha ERS. As a control analysis it would be interesting to check if the maximal effect was found over the selected channel or in the left side. The left side seems to show even a higher activation than in the right selected channel. And, if this is the case, I would repeat the analysis using left and right channels just as it has been done for the ERD analyses.  

Figure 2, 4 and 5. It would be advisable to present these data with a violin plot or similar in order to provide information about the distribution of the data.

Figure 3. In the TFRs plots from panel A, I would be helpful to present a dashed line with the stimulus onset/offset for encoding and probe stages to help the reader to understand the dynamics of the oscillations during the task. 

Figure 5. Typo in the X axis: "difficulty"

Please report effect sizes throughout. 

Discussion

In general, the results are very well in line with work from working memory n-back task. Can the authors comment presente findings from the working memory and oscillations literature?

439-434: There are several statements related to the functional meaning of alpha desyn/synchronization. Please, provide a reference after each statement.

484: I found this part of the discussion very interesting and this is just a comment, I don't need further discussion in the paper about this. As the authors, I was also surprised by the fact that the ERD during the probe was especially modulated by the the most difficult conditions. The authors argued that this could happen because contrary to the classical notions of effort, effort may be a process-specific phenomenon. Thinking about the nature of the task I think that the easier conditions (3,4 items) and the harder conditions (5,6 items) may be recruiting different mnemonic processes. The participants may have been able to solve the easier conditions just by familiarity whereas the harder conditions may have been required recollection processes. Familiarity is almost automatic whereas recollection is a more effortful process and that could be the reason why the differences emerged only when more items should be evaluated (5,6 items). In other words, effort would be a general phenomenon but in the easier conditions of the task one can't evaluate it because familiarity does not require effort. In the future, it would be very interesting to disentangle these effects related to effort and task stages.

Author Response

We appreciate the positive comments and constructive suggestions for improving our article. Below we address each comment and made corresponding edits in the text. Please note that the line numbers correspond to the tracked version of the text. 

Introduction: 

  1. The authors nicely present some of the classical views of the functional role of alpha synchronization and desynchronization. However, in my opinion a complete picture of the current knowledge of this functional role is missing because some seminal works related to alpha are not presented in the Introduction. For example, the role of alpha related to the inhibition of task-irrevelant areas (Jensen et al, 2010), idling (Pfurtscheller and Lopes da Silva, 1999) or the amount of information retrieved (Hanslmayr et al., 2012, 2016; Griffiths et al., 2019). 

Thank you for pointing it out. We agree that research by Jensen et al., (2010), Pfurtscheller and Lopes da Silva, (1999), Hanslmayr et al., (2012, 2016) and Griffiths et al., (2019) have been fundamental for understanding the role of alpha synchronization and desynchronization. We have revised the text to add these references in lines 105-107: “Some authors suggest that alpha desynchronization reflects the amount of encoded and retrieved information that is parametrically related to the decrease of synchronized firing of neurons [48-50]“

  1. Also, the exact functional significance of alpha oscillations is still a matter of debate and in my opinion, this should be noted in the Introduction. 

Thank you for the comment. Indeed, there are different opinions about the functional significance of alpha oscillations, including its role as a biological artifact, marker of cognitive inactivity, correlate of inhibitory control and redistribution of neural resources. We have improved the narrative to indicate this matter in lines 82-87: “Different opinions exist about the functional significance of the alpha rhythm. Some authors suggest that EEG alpha oscillations do not have any functional significance and reflect a biological artifact [28], whereas others propose that the alpha-rhythm is a marker of cognitive inactivity or generalized idling brain state [29, 30]. More recent hypotheses relate the alpha rhythm with inhibitory control and redistribution of neural resources [31].  “

  1. The authors examined the alpha correlates (ERD and ERS) found with the Stenberg task but they didn't relate them to the functional meaning of these correlates. Here I miss a brief explanation of the modulation of the alpha band in function of the memory load, distractors, memory performance, cognitive effort... In this study I found especially important the memory load modulation.

Thank you for the thoughtful comment. In the text we point to research that examined alpha correlates to cognitive task and some that modulated task complexity in terms of memory (e.g., 8, 19, 45) and in terms of managing distractors (e.g., 51). We elaborate on these studies to highlight that alpha power modulations appear to be driven by memory load as they require an increased need for sustained attention and reflected by an increase in the amount of allocated cortical resources. Further, the modulation of alpha synchronization may be explained through the increase of top-down functional inhibition of regions that can possibly perturb working memory maintenance. These edits are on lines 102-107: “This effect reveals the relation of alpha desynchronization with attentional demands and may reflect an increase in sustained attention and in the amount of allocated cortical resources [40], which may be driven by activation of thalamocortical networks associated with memory processes [47]. Some authors suggest that alpha desynchronization reflects the amount of encoded and retrieved information that is parametrically related to the decrease of synchronized firing of neurons [48-50].“

Data analysis:

  1. Line 255: Crucial parameters for time-frequency decomposition are missing: the width of the wavelets in number of cycles, frequency band of interest, times on which the analysis windows was centered. 

Thank you for the comment. Initially we had mentioned parameters on frequency band of interest later in the text (lines 270-271 in original article: “For statistical analysis, individual alpha frequency was used (individual peak of oscillations power in the frequency range from 8 to 13 Hz).”). Following your suggestion, we added information about width of wavelets in number of cycles and times on which the analysis windows was centered to lines 299-303 in the revised article. “The 7000 ms were used with window centered at each 1000 ms time step from 1 to 40 Hz. A set of good tapers (i.e., those with the least leakage from far away frequencies) were chosen automatically based on MNE algorithms. The width of wavelets was different for different bands and was chosen so that the number of cycles was equal to frequency/2, where frequency was a range from 1 to 40 Hz.”

  1. Line 263: Please, provide the final number of clean trials per condition (mean and SD) that were used in the subsequent analyses.

Thank you. We have added information about the final number of trials per difficulty conditions in the lines 289-292: “After removing artifacts, the following number of trials remained in the data: mean = 25±1 for the 3-digit condition, mean = 34±2 for the 4-digit condition, mean = 28±2 for the 5-digit condition, and mean = 32±3 for the 6-digit condition.”

  1. Line 283: Please, provide information about the amount of data discarded.

Thank you. We have added information about amount of removed outliers for each variable in lines 328-332: “The following values were removed as outliers for each variable: 1 value for the encoding-related ERD variable, 0 values for the retention-related ERS variable, 1 value for the probe-related ERD variable, 9 values for the reaction time variable, 7 values for the accuracy variable, 4 values for the Raven score variable, 4 values for the mental toughness score variable .”

  1. Line 292: This was already stated before.

Thank you. We have eliminated the extra sentence about only correct responses.

Results:

  1. The authors do an a priori selection of the channels of interest based on previous literature. However, the topoplot of Figure 3B related to the maintenance stage shows a bilateral alpha ERS. As a control analysis it would be interesting to check if the maximal effect was found over the selected channel or in the left side. The left side seems to show even a higher activation than in the right selected channel. And, if this is the case, I would repeat the analysis using left and right channels just as it has been done for the ERD analyses.  

Thank you for pointing this out. In fact, we used two electrodes for the retention-related ERS analysis – PO8 and PO7 combined. The methods erroneously stated that only the right electrode was used. The error was due to the fact that there are past studies demonstrating a rather right lateralization of the effect, and we should have clarified that although past studies sometimes show a right lateralization, we have used electrodes from both hemispheres, as we had stated in the discussion (lines 458-460 in the original version: “An alternative explanation may be that the ERD effect related to encoding overlapped the ERS effect, since the ERD preceded ERS and affected electrodes used for ERS analysis (PO7-PO8).”). We take the opportunity to clarify this in the method section. Further, to verify whether electrodes in the left (PO7) and right (PO8) hemispheres yielded comparable results, we compared activity from electrodes PO8 and PO7 using mixed models with alpha ERS activation as a target variable, electrode as a predictor, subject ID as a random factor. We also repeated this analysis for each task difficulty level separately. All results showed no significant differences between the electrodes (p-value > 0.05), confirming that we can combine activity from both electrodes. We clarify that we combined data from PO8 and PO7 and mention results of this analyses in the current revision.

Lines 314-316 “whereas alpha rhythm activation was expected to be observed on the parietal-occipital lateral electrodes (PO7, PO8), in accordance with previous EEG studies using the Sternberg task [20, 56].“

  1. Figure 2, 4 and 5. It would be advisable to present these data with a violin plot or similar in order to provide information about the distribution of the data.

Thank you for the comment. We have replaced Figures 2, 4 and 5 with plots showing data distribution, means and standard errors (using the R pirateplot function).

  1. Figure 3. In the TFRs plots from panel A, I would be helpful to present a dashed line with the stimulus onset/offset for encoding and probe stages to help the reader to understand the dynamics of the oscillations during the task. 

Thank you. We have highlighted time of stimuli presentation in the TFRs plots and added this information to the figures’ description.

  1. Figure 5. Typo in the X axis: "difficulty".

Thank you, we have corrected the typo.

  1. Please report effect sizes throughout. 

Thank you. Because we have used a mixed model analyses, we have implemented effect size estimates for multilevel models as suggested by Lorah (2018). Specifically, effect sizes were calculated using the following formula: f2 = (R22 – R12)/(1- R22), where R22 represents the variance explained for a model with the given effect and R12 represents the variance explained for a model without the given effect (Lorah, 2018, https://doi.org/10.1186/s40536-018-0061-2). Please note that this measure reflects the proportion of variance explained by the given effect relative to the proportion of outcome variance unexplained and is considered small at a value of 0.02, medium at a value of 0.15, and large at a value of 0.35 (Cohen, 1992, https://doi.org/10.1037/0033-2909.112.1.155).  We have added f2 scores in the Results section ( lines 405-467). The description of the effect size calculation has also been added in the Methods section , lines 364-369. “For each fixed effect of each model, the effect size was calculated using the following formula: f2 = (R22 – R12)/(1- R22), where R22 represents the variance explained for a model with the given effect and R12 represents the variance explained for a model without the given effect [81]. This measure reflects the proportion of variance explained by the given effect relative to the proportion of outcome variance unexplained and is considered small at a value of 0.02, medium at a value of 0.15, and large at a value of 0.35 [82].”

Discussion

  1. In general, the results are very well in line with work from working memory n-back task. Can the authors comment presente findings from the working memory and oscillations literature?

Thank you. Indeed, the n-back task is similar to the Sternberg task in some respects as it requires cognitive processes such as encoding, retention that involves inhibition of irrelevant information, and retrieving. These cognitive processes should be represented by the same oscillatory effects. Critically, it is important to point out that the n-back task requires simultaneous activation of these processes. Encoding is reflected by alpha desynchronization, while retention - by alpha synchronization. Therefore, the n-back task does not lend itself for studying oscillatory correlates of these processes separately as they are cooccur during the task (Klimesch et al., 2005, DOI: 10.1027/1618-3169.52.2.99). We have improved the narrative to discuss this point on lines 582-595: “Critically, the alpha ERD effect that shows increases with task difficulty is also observed in studies using other working memory tasks, for example, the n-back task [13, 14, 42, 45, 55]. However, studies using the n-back task do not report results associated with a alpha ERS effect. Although the Sternberg task is similar to the n-back task in requiring encoding, retention, and retrieving, the n-back task requires simultaneous implication of these processes, which leads to an overlapping in some of their neural correlates [55]. Therefore, the increasing alpha ERD with task difficulty can be associated with both encoding and retrieving, and can also be distorted by the overlapping ERS effect. Interestingly, Fairclough and Ewing [14] observed increase of absolute alpha power in the most difficult 7-back condition. This effect may indicate the drop of effort in the very difficult condition, as described by the Motivational Intensity theory. We suggest that a study using the Sternberg task that includes very difficult conditions and ERD/ERS measures may more clearly reveal this effect as a stimulus-related correlate of task performance and show which specific process this effect refers to. ”

  1. 439-434: There are several statements related to the functional meaning of alpha desyn/synchronization. Please, provide a reference after each statement.

Thank you. We have added references after each statement about the functional meaning of alpha (de)synchronization (lines 506-511).

  1. 484: I found this part of the discussion very interesting and this is just a comment, I don't need further discussion in the paper about this. As the authors, I was also surprised by the fact that the ERD during the probe was especially modulated by the the most difficult conditions. The authors argued that this could happen because contrary to the classical notions of effort, effort may be a process-specific phenomenon. Thinking about the nature of the task I think that the easier conditions (3,4 items) and the harder conditions (5,6 items) may be recruiting different mnemonic processes. The participants may have been able to solve the easier conditions just by familiarity whereas the harder conditions may have been required recollection processes. Familiarity is almost automatic whereas recollection is a more effortful process and that could be the reason why the differences emerged only when more items should be evaluated (5,6 items). In other words, effort would be a general phenomenon but in the easier conditions of the task one can't evaluate it because familiarity does not require effort. In the future, it would be very interesting to disentangle these effects related to effort and task stages.

Thanks a lot for this interesting comment! The idea about automatic processes did not come to mind and it is very interesting. We agree that the effect of familiarity with the stimulus may explain the effect of alpha desynchronization increase in 5-6-digits condition in participants with low mental toughness. However, we also would like to point out that there was no alpha desynchronization increase with task difficulty in participants with high mental toughness and no mental toughness effect was observed for the encoding and retention phases. In our opinion, this may be one of the reasons our results support the notion that effort is process-specific. However, we are aware that this is the first study to report this phenomenon in using the Sternberg task and it is important for future studies to replicate this effect and disentangle effects related to effort and task stages.

Round 2

Reviewer 1 Report

Good job for the authors.

Reviewer 2 Report

Thanks for the effort to respond to my comments. The quality of the paper has significantly increased and it meets all the standards of the journal.